# Effects of High-Intensity Interval Training Using the 3/7 Resistance Training Method on Metabolic Stress in People with Heart Failure and Coronary Artery Disease: A Randomized Cross-Over Study

**DOI:** 10.3390/jcm12247743

**Published:** 2023-12-17

**Authors:** Alexis Gillet, Kevin Forton, Michel Lamotte, Francesca Macera, Ana Roussoulières, Pauline Louis, Malko Ibrahim, Céline Dewachter, Philippe van de Borne, Gaël Deboeck

**Affiliations:** 1Department of Cardiology, CUB Hôpital Erasme, Hôpital Universitaire de Bruxelles (H.U.B), Université Libre de Bruxelles (ULB), 1050 Brussels, Belgium; alexis.gillet@ulb.be (A.G.); kevin.forton@ulb.be (K.F.); michel.lamotte@hubruxelles.be (M.L.); francesca.macera@ulb.be (F.M.); ana.roussoulieres@hubruxelles.be (A.R.); celine.dewachter@hubruxelles.be (C.D.); philippe.van.de.borne@ulb.be (P.v.d.B.); 2Department of Physiotherapy, CUB Hôpital Erasme, Hôpital Universitaire de Bruxelles (H.U.B), Université Libre de Bruxelles (ULB), 1050 Brussels, Belgium; pauline.louis@hubruxelles.be; 3Research Unit in Rehabilitation Sciences, Faculty of Motor Skills Science, Université Libre de Bruxelles, 1070 Brussels, Belgium; malko.ibrahim@ulb.be; 4Laboratory of Physiology and Pharmacology, Faculty of Medicine, Université Libre de Bruxelles, 1070 Brussels, Belgium

**Keywords:** strength training, HIIT, cardiac rehabilitation, cardiovascular disease

## Abstract

The 3/7 resistance training (RT) method involves performing sets with increasing numbers of repetitions, and shorter rest periods than the 3x9 method. Therefore, it could induce more metabolic stress in people with heart failure with reduced ejection fraction (HFrEF) or coronary artery disease (CAD). This randomized cross-over study tested this hypothesis. Eleven individuals with HFrEF and thirteen with CAD performed high-intensity interval training (HIIT) for 30 min, followed by 3x9 or 3/7 RT according to group allocation. pH, HCO^3−^, lactate, and growth hormone were measured at baseline, after HIIT, and after RT. pH and HCO^3−^ decreased, and lactate increased after both RT methods. In the CAD group, lactate increased more (6.99 ± 2.37 vs. 9.20 ± 3.57 mmol/L, *p* = 0.025), pH tended to decrease more (7.29 ± 0.06 vs. 7.33 ± 0.04, *p* = 0.060), and HCO^3−^ decreased more (18.6 ± 3.1 vs. 21.1 ± 2.5 mmol/L, *p* = 0.004) after 3/7 than 3x9 RT. In the HFrEF group, lactate, pH, and HCO^3−^ concentrations did not differ between RT methods (all *p* > 0.248). RT did not increase growth hormone in either patient group. In conclusion, the 3/7 RT method induced more metabolic stress than the 3x9 method in people with CAD but not HFrEF.

## 1. Introduction

The central strategy of secondary cardiovascular disease (CVD) prevention is cardiovascular rehabilitation [1,2,3,4,5]. Therefore, multidisciplinary CR is a class 1A intervention that should be offered to every individual with CVD. One of the specific core components of CR is exercise training, including aerobic and resistance training (RT) [2,6,7].

High-intensity interval (aerobic) training (HIIT) leads to greater improvements in aerobic capacity and cardiac function in people with heart failure and CVD than moderate continuous training [7,8,9]. HIIT increases blood lactate concentration and other metabolic byproducts that have been shown to increase aerobic capacity and mitochondrial biogenesis in skeletal muscle [9,10].

RT improves exercise capacity [6,11,12] and quality of life and reduces hospitalization rates in both people with heart failure with reduced ejection fraction (HFrEF) [2,13,14] and those with coronary artery disease (CAD) [15]. It has also been shown to reduce cardiovascular mortality in people with CAD [15]. RT prescription is advocated for people with CVD, although the most effective strategy is still debated [11,16,17,18,19]. Recent guidelines recommend the use of the 3x9 method, which consists of 3 sets of 9 repetitions (27 repetitions in total) at 70% of the one-maximal repetition (1-RM) with a 60 s recovery time between successive sets [6,20,21,22] (Appendix A).

Methods to increase muscle strength are classically based on the “overload” principle [23]. Moderate to high mechanical muscle loading (≥60–70% of 1-RM) has long been considered as the main stimulus (i.e., mechanical tension) for muscle hypertrophy and increased muscle strength [20]. However, studies suggest that the accumulation of fatigue-related metabolites (i.e., metabolic stress) may play a complementary role in the exercise stimulus, leading to an increased accretion of muscle mass and strength [24,25]. This is particularly interesting because metabolic stress has been shown to stimulate skeletal muscle hypertrophy [25], mitochondrial biogenesis [26], and angiogenesis [27]. This metabolic stress, which is commonly measured by blood lactate concentration [28], also increases during RT [29,30]. Initial studies on adaptive hypertrophy primarily examined the temporary increase in anabolic hormones in the blood after exercise, including growth hormone and cortisol [31,32,33,34]. This led to an understanding of the activation of a specific signaling pathway involving phosphatidylinositol 3-kinases due to the interaction of insulin-like growth factor with insulin and insulin-like growth factor receptors [31]. The number of repetitions in a set determines the time under tension of the muscle, which is directly related to blood lactate concentration after RT [35,36,37,38].

The 3/7 RT method requires less time than the 3x9 method and has been shown to effectively increase muscle strength in healthy individuals [39,40]. This method involves performing five sets, each separated by only 15 s, with an incremental number of repetitions per set (3 repetitions, 4 repetitions, 5 repetitions, 6 repetitions, and 7 repetitions: 25 repetitions in total) at a constant load of ~70% of 1-RM [39] (Appendix A). A study using near-infrared spectroscopy technology in physically active individuals showed reduced O_2_ delivery to the active muscles during the 3/7 method [41]. This was accompanied by an increase in metabolic stress, increased strength, and neuromuscular adaptation [40,41,42].

We recently showed that the 3/7 method induced a safe, submaximal hemodynamic response (heart rate, stroke volume, and blood pressure) in people with HFrEF and CAD. Furthermore, in people with HFrEF, the hemodynamic response to the 3/7 and 3x9 RT methods was similar, whereas it was higher for the 3/7 method in the CAD group only [43].

This randomized, cross-over study aimed to test the hypothesis that the 3/7 method induces more metabolic stress in individuals with HFrEF and CAD than the 3x9 method when associated with HIIT.

## 2. Materials and Methods

Study design: A single-center, randomized, crossover study was conducted at the cardiovascular center of the “Erasme University hospital” in Brussels. The protocol was approved by the local Research Ethics Board and registered on ClinicalTrials.gov Identifier: NCT05391620. All participants provided written informed consent; there was no financial compensation for participation.

Participants were randomly allocated using a computer-generated allocation schedule to perform either the RT 3x9 method on day 1 and RT 3/7 on day 2 or the opposite, with a minimum rest period of two days, although a 7-day rest was considered ideal [44]. Participants and evaluators were not blinded to the RT method performed.

### 2.1. Participants

Consecutive men with stable HFrEF or CAD who were participating or had previously participated in an exercise program were recruited by the principal investigator. Participants with HFrEF attributed to CAD were analyzed in the HFrEF group. The stability of HFrEF was defined by a left ventricle ejection fraction (EF) ≤ 40% for more than 3 months and stabilized by a maximally tolerated HFrEF treatment recommended by the latest guidelines for the management of heart failure [2]. Stability of CAD was defined as more than 1 uneventful month since an acute coronary syndrome and/or percutaneous coronary intervention if left ventricular EF was ≥50% (Table 1). Participants had to be >18 years old, with no limiting orthopedic or neurologic disorders. Exclusion criteria were symptomatic heart valve disease and signs of cardiovascular instability, such as angina, and/or electrocardiographic evidence of myocardial ischemia during exercise.

All participants performed a cardiopulmonary exercise test (CPET) and a strength test 1 week before participation in the protocol. HIIT was tailored to each individual according to their CPET results as recommended [2,6]. During CPET (see Appendix B), hemodynamic variables such as heart rate and blood pressure were measured at rest, and throughout the effort, anaerobic threshold was determined by the V-slope method and the VE/VCO_2_ slope utile the second ventilatory threshold [45].

### 2.2. Outcomes

All measurements were performed by the same physiotherapist. Measurements were performed during 2 training sessions at the same time of day (between 1 and 3 p.m.) and separated with a minimum rest period of two days, although a 7-day rest was considered ideal [44]. Each participant was instructed to refrain from any strenuous activities the day before. Blood samples and ratings of perceived exertion (RPE) were taken at rest before training (t0), directly after the HIIT (t1), and 2 min after the end of the RT (t2) (Figure 1).

#### 2.2.1. Blood Samples

Venous blood samples (10 mL) were drawn from the median cubital vein using a catheter. Two vacutainers of blood samples were drawn (Becton Dickinson, San Jose, CA, USA); the first contained EDTA for plasma separation, and the second was a heparinized syringe for using cartridge-based technology. The samples were centrifuged at 2500 rpm for 15 min with plasma and serum aliquots then stored at −280 °C until analysis. Cortisol (total) concentration was analyzed by enzyme-linked immunoassays (DSL, Austin, TX, USA) following the manufacturer’s protocols. The detection limit of the cortisol assay was <0.05 nmol·L^−1^ with intraassay and interassay CVs of 4.1% and 9.8%, respectively. We used GEM Premier 5000 (Zavetem, Belgium) with cartridge-based technology for the second vacutainers; each cartridge contained sensors to measure pH, HCO^3−^, and lactate.

#### 2.2.2. RPE

RPE during RT was assessed using the modified Borg Scale (0–10). To determine if participants preferred one RT method over the other, we asked them to rate how much they enjoyed the RT method on a scale from 0 to 10, where 0 = “strongly disliked” and 10 = “very much enjoyed” at t2.

### 2.3. Interventions

The training session was composed of 30 min of HIIT followed by a session of either the 3/7 RT method (14 min) or the 3x9 RT method (19 min) according to group allocation. (Figure 1).

#### 2.3.1. High-Intensity Interval Training

The 30 min HIIT consisted of a 6 min warm-up at 50% of peak power output followed by six blocks of 2 min work periods at 80% of peak power output interspersed with 2 min of active recovery on an upright cycle ergometer (Monark©-ergomedic 828E). Heart rate was recorded at the end of each interval using an HR monitor (HR 300, Decathlon, Lille, France), which was worn around the chest during the whole training session. The distance at the end of the HIIT was recorded.

#### 2.3.2. Resistance Training

Both RT methods were performed using a 10 RM load determined for each RT exercise using the usual standardized method prior to the first training session. Participants were taught to perform the exercises correctly and were asked to lift each load via their full range of motion without Valsalva.

The 3x9 method consisted of 3 sets of 9 repetitions (27 repetitions in total) with a 60 s recovery between successive sets. The 3/7 method consisted of 5 sets with an increase from 3 to 7 repetitions (total 25 repetitions), separated by periods of 15 s recovery. The exercises involved consecutive concentric and eccentric contractions at a cadence that was identical for both methods (i.e., 1 s/1 s) (Appendix A).

The order of RT was leg press, dips machine, seated leg curl, vertical traction, and leg extension machines, and the time to change to the next machine was one minute. The same investigator supervised each workout to ensure the correct technique was used and provided verbal encouragement.

### 2.4. Statistical Analyses

The effects of method and time on pH, lactate, and HCO^3−^ concentrations were evaluated using linear mixed-effects models. Time was considered nested within methods. The analyses were performed separately for each group (HFrEF and CAD). HIIT performed before RT 3x9 and before RT 3/7 was also analyzed separately. We verified the presence of a carry-over effect by analyzing the interaction between time and group (3/7 RT followed by 3x9 RT vs. 3x9 RT followed by 3/7 RT).

Differences between time points were analyzed using the least squares method with a *p*-value adjusted by the Tukey method. The influence of different parameters on these measures was also analyzed using linear mixed-effects models. The normality of residuals was checked using graphical representations (histograms and boxplots). The significance level was set at 0.05. The ratings of enjoyment of each RT method and the RPE were compared between groups using the Mann–Whitney–Wilcoxon test. Differences in CPET results were analyzed using Student’s *t*-test or Mann–Whitney–Wilcoxon test as appropriate. The analyses were conducted using SAS Enterprise Guide 9.3 software.

## 3. Results

Eleven participants with HFrEF and fourteen with CAD were enrolled. None reported any discomfort during the training, and no adverse events occurred. The number of days between the two tests was similar (7 ± 4 for HfrEF and 9 ± 4 days for CAD (*p*= 0.140)). The mean ages of the HfrEF and CAD groups were 59 ± 17 and 61 ± 13 years, respectively. All demographic and clinical characteristics of participants are presented in Table 1.

Mean peak oxygen consumption was similar in both groups (HfrEF: 20.7 ± 7.1 and CAD: 21.3 ± 4.8 mL/min.Kg, *p* = 0.601). The CPET profiles of both groups are presented in Table 2.

### 3.1. Metabolic Effects in HfrEF 

For the HfrEF group, pH was not affected by HIIT but decreased similarly after both RT modalities. Bicarbonate ion (HCO^3−^) fell after HIIT and even further so after 3/7 RT, while the reduction after 3x9 RT was not significant. HCO^3−^ did not differ after RT modalities. Lactate increased with HIIT and even more thereafter with RT, and in a comparable manner with both RT modalities (Figure 2).

### 3.2. Metabolic Effects in CAD

For the CAD group, pH was not affected by HIIT but decreased similarly after both RT modalities, with a tendency for a lower pH after the 3/7 RT (*p* = 0.060). HCO^3−^ fell after HIIT with no further change after both RT modalities. However, HCO^3−^ was lower after 3/7 RT than 3x9 RT (*p* = 0.004). Lactate concentration increased after HIIT, with a further increase after 3/7 RT but not after 3x9 RT. The 3/7 method induced a higher lactate concentration than the 3x9 method at t2. (Figure 2).

### 3.3. Hormonal Effect in HFrEF

In the HFrEF group, growth hormone concentration did not change after HIIT or after RT, and there were no differences between training sessions involving the RT 3/7 or 3x9 method. Cortisol concentration also stayed unchanged during the training sessions. (Table 3).

### 3.4. Hormonal Effect on CAD

In the CAD group, growth hormone concentration increased with no further increase after RT. Cortisol concentration stayed unchanged during the training sessions (Table 3).

### 3.5. Rating of Perceived Exertion

In the HFrEF group, there was no difference between the mean RPE ratings for RT 3/7 and 3x9, suggesting a similar level of perceived difficulty. There was a tendency toward a preference for the 3/7 RT method (Table 4).

In the CAD group, there was no difference between the mean RPE ratings for RT 3/7 and 3x9. There was no preference for either type of training (Table 4).

No participants dropped out of the study, but two training sessions had to be rescheduled because of technical issues with the catheter for the measurement of blood concentrations.

## 4. Discussion

This study partially confirmed the hypothesis that the 3/7 RT method would induce more metabolic stress than the 3x9 method after a HIIT session, specifically in participants with CAD. However, this was not observed in the HFrEF group.

In the CAD group, the 3/7 method increased lactate and reduced HCO^3−^ concentrations and tended to lower pH more than the 3x9 method. However, the kinetics of the pH and lactate and HCO^3−^ concentrations were similar between methods in both groups.

The 3/7 method is a new RT modality [42] that induces a reduction in O_2_ supply during and after muscle work by using a short recovery period (i.e., 15 s). In healthy individuals this causes an ischemic/hypoxic condition in the muscle cells, increasing the dependency on anaerobic metabolism and reducing the possibility to restore muscle homeostasis [41]. This method has been shown to alter ionic concentration gradients (K^+^, Na^+^, and Ca^2+^) and to impact the accumulation of metabolic byproducts, including lactate, hydrogen ions, inorganic phosphate, adenosine diphosphate, and others. All these factors have been proposed to play a role in maximal strength and muscle hypertrophy [40,46,47].

As stated above, participants with CAD and HFrEF responded differently to each type of RT. One reason for this difference could be the chronic myopathy often present in the case of HFrEF [48]. Qualitative and quantitative changes, such as muscle wasting/cachexia, a shift from slow (fatigue resistant) to fast (fatigue non-resistant) fiber type, and reduction in mitochondrial density and enzymes, associated with inflammatory status and neurohumoral changes might cause reduced muscle force, early muscle fatigue, and decreased aerobic capacity [49,50]. This chronic muscle failure and lack of enzyme dynamics could explain the slightly lower lactate level after HIIT in HFrEF than in CAD participants. Although both pathology groups trained at the same intensity, the participants with HFrEF appeared to be in a situation of metabolic failure that prevented them from producing higher metabolic stress and limiting muscle homeostasis recovery by the usual cellular pathways. This might be explained by a slower rate of oxygen use after an anaerobic threshold in HFrEF [51].

However, despite typical abnormalities in the CPET profile of the participants with HFrEF (i.e., high VE/VCO_2_), the aerobic capacity of the HFrEF and CAD participants was similar. Indeed, oxygen consumption at the anaerobic threshold and at peak exercise was similar for both groups, with normal VO_2_/WR slopes that were probably related to normal muscle oxygen use during the exercise. Moreover, the HFrEF group had a mean VO_2_p of 20 mL/kg/min, which is unlikely to be associated with muscle abnormalities commonly seen in highly deconditioned people with HFrEF [52,53]. Therefore, we believe that chronic myopathy might only explain a small part of the different metabolic stress responses to exercise between the two groups.

Iron deficiency could also explain different metabolic stress responses during exercise in the muscle cells of people with HFrEF. People with HFrEF and iron deficiency have lower peak muscle strength, higher levels of energy depletion, and more pronounced muscle acidification during exercise, consistent with a metabolic shift to anaerobic pathways [54] and rapid metabolic failure. However, this explanation seems unlikely as only 1 participant with HFrEF had an iron deficiency and, although his exercise capacity was lower than that of the other participants in the HFrEF group (VO_2_p 17 mL/kg·min), his metabolic response to HIIT and RT did not differ from that of the other participants. In addition, two participants with CAD had iron deficiency but had unremarkable exercise capacities and metabolic responses during both exercise sessions (VO_2_p 27 and 20 mL/kg·min).

We believe that the most likely explanation for the different metabolite concentrations between the groups is the different cardiac output adaptation capacities during exercise in the case of HFrEF and CAD [2]. Lower cardiac output could reduce oxygen transport to exercising muscles and/or poor metabolite clearance from the muscle cells, although adaptation to RT is not specifically limited or altered by cardiac output in people with HFrEF [55]. However, in a previous study, we found lower hemodynamic adaptation during RT (using the same methods as those used here) in people with HFrEF than those with CAD, although a similar cardiac output was achieved at the end of both the 3/7 and 3x9 methods in the HFrEF group [43]. We did not measure hemodynamic adaptation in the present study. However, a faster reacting cardiovascular system in people with CAD than HFrEF might be the reason for the higher metabolite concentration in the participants with CAD because of better energy supply to the muscle cells allowing more stable homeostasis [56]. Because muscles reach their maximal capacity in both the 3x9 and 3/7 RT methods, there is no reserve for further increments in HFrEF. However, people with CAD may have similar cardiovascular adaptation to people without any heart pathology because of a normally reactive but deconditioned system [41,43]. The 15 s recovery period might therefore be too short to repay the muscle oxygen debt and perform muscle cell clearance in people with HFrEF, even if recovery periods of 60 s were allowed between RT machines. Too-short recovery periods (i.e., 15 s) are known to trigger a central vasoconstriction reflex, and it is well known that HFrEF is associated with altered chemo/ergo reflex and central reflex dysregulation demonstrated by the high VE/VCO_2_ ratio and steep VE/VCO_2_ slope [57,58]. We therefore believe that inadequate cardiovascular adaptation during RT in HFrEF might explain our results [59].

Several studies have shown that HIIT improves VO_2_p and VO_2_ at an anaerobic threshold more than continuous training in healthy individuals [9,60] and in people with CAD [8]. However, results are contradictory in HFrEF [61,62], where some studies have shown that it is more difficult to ensure that target intensities are achieved. The “Smartex study” showed that VO_2_p improved similarly in people with HFrEF because they finally trained at an equivalent mean heart rate during HIIT and continuous training [63]. We systematically and carefully ensured that all participants could reach the target training heart rate by providing HIIT for only 2 min, in contrast with the SMARTEX study that used 4 min of high intensity training. This shorter duration of HIIT might be a good alternative for people with HFrEF with limited cardiac output response to exercise and who need to pace their effort for HIIT that lasts for 4 min [6].

However, the time course of the metabolites was similar between the HFrEF and CAD groups during HIIT. It is known that H^+^ ions, and therefore anaerobic training, trigger the chemoreceptors and mediate the release of growth hormone by the anterior pituitary gland [64,65]. pH and lactate concentration were substantially modified by the exercise in both groups, as in healthy subjects [66]. It is therefore surprising that growth hormone concentration only increased in the CAD but not the HFrEF group. Neurohumoral dysfunction is a key feature of HFrEF [2], but very little information is available regarding the pituitary gland secretion of growth hormone in HFrEF. Importantly, it has been shown recently that about a third of people with HFrEF have a growth hormone deficiency associated with a poor functional and hemodynamic status [67]. In our study, it was not possible to link any change in growth hormone concentration with functional or hemodynamic status.

Regarding exercise preference, the participants performed high-intensity RT at ~70% 1-RM, which corresponds to the third step in cardiac rehabilitation [7]. All the participants successfully completed the exercises within the time constraints (i.e., 15 s of rest between sets) without external help; the perception of effort can be considered as “hard” for the HFrEF and “really hard” for the CAD group. Therefore, the use of the 3/7 method as part of CR is feasible and does not generate a greater perception of effort compared with the third step of the CR guideline. For optimal adherence to exercise prescription, individuals need to understand the benefits of exercise, and exercise should be individualized according to their preferences [7]. Our study shows that the participants had a small preference for the 3/7 RT method rather than the 3x9 method after HIIT.

### Limitations

Our study is based on a relatively small number of patients with a large age range and with a typical population of European cardiac centers. However, we excluded women due to fluctuations in female sex hormones during the menstrual cycle that may affect exercise training response [68]. Furthermore, HFrEF and CAD demonstrate a higher participation rate in CR in males [11]. Given the small sample sizes, proposing a sufficiently large subgroup of females was not feasible in our center; we did not include them, and this warrants further investigation. Whether the acute effects reported in this study will translate into long-term beneficial effects is unknown and will require further long-term studies. And we included people who were used to performing activity. It would be interesting to verify the results in people at the start or end of a cardiac rehabilitation program since training could affect the results.

## 5. Conclusions

This study showed that the 3/7 RT method induced more metabolic stress than the 3x9 method after a HIIT session in people with CAD, but this was not observed in the HFrEF. Therefore, we suggest that the 3/7 RT method could be used as part of a cardiac rehabilitation program to improve muscle strength in people with CAD.

## Figures and Tables

**Figure 1 jcm-12-07743-f001:**
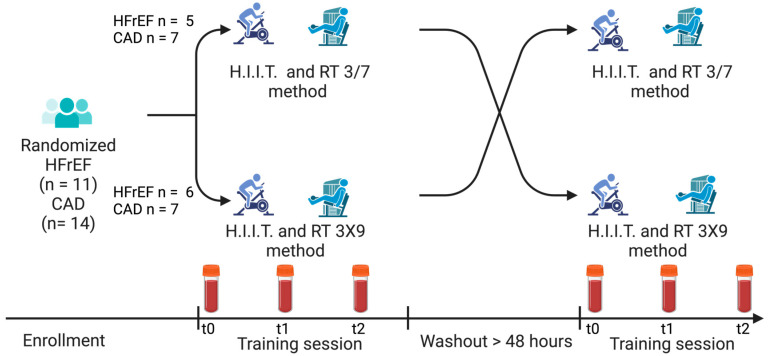
Description of the study schedule. t0 was at rest, t1 was 2 min after high-intensity interval training (H.I.I.T.), and t2 was 2 min after resistance training (RT). CAD, coronary artery disease; HFrEF, heart failure with reduced ejection fraction (≤40%).

**Figure 2 jcm-12-07743-f002:**
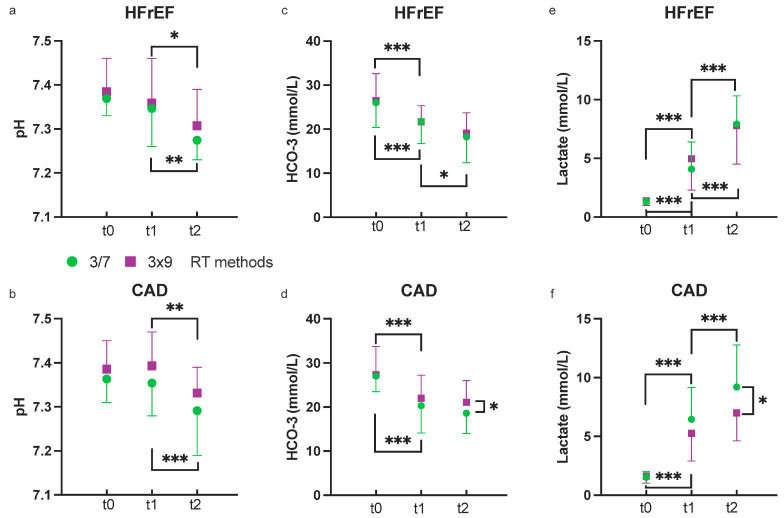
Change in pH, HCO^3−^, and lactate during HIIT (t1) and 3x9 (purple squares) and 3/7 RT (green circles) in participants with HFrEF (**a**,**c**,**e**) or CAD (**b**,**d**,**f**); CAD, coronary artery disease; HFrEF, heart failure with reduced ejection fraction (<40%) * *p* < 0.05 ** *p* < 0.01 *** *p* < 0.001.

**Table 1 jcm-12-07743-t001:** Demographic and Clinical characteristics of participants with heart failure and coronary artery disease.

Characteristic	HFrEF *n* = 11	CAD *n* = 14	*p*-Value
Age (years)	59 ± 17	61 ± 13	0.238
Weight (kg)	83 ± 21	86 ± 13	0.428
Height (cm)	172 ± 8	175 ± 3	0.428
BMI (kg/m^2^)	28.06 ± 6.84	28.02 ± 3.43	0.428
Diabetes mellitus, *n* (%)	2 (18)	2 (14)	0.796
Smoking, *n* (%)	3 (27)	6 (43)	0.442
EF < 40%, *n* (%)	11 (100)	-	<0.001
Heart failure caused by ischemic heart disease, *n* (%)	5 (45)	-	-
Antiplatelet agents, *n* (%)	8 (73)	13 (93)	0.182
Statins, *n* (%)	8 (73)	14 (100)	0.041
β-Adrenergic antagonists, *n* (%)	11 (100)	11 (79)	0.109
Diuretics, *n* (%)	10 (91)	1 (7)	<0.001
ACE inhibitors, *n* (%)	3 (27)	7 (50)	0.337
Angiotensin II receptor antagonists, *n* (%)	2 (18)	1 (7)	0.358
Sacubitril/valsartan, *n* (%)	8 (73)	-	<0.001
Empaglifozine/dapagliflozin, *n* (%)	4 (36)	1 (7)	0.076
Previous CABG, *n* (%)	2 (18)	2 (14)	0.796
Previous PCI, *n* (%)	3 (27)	12 (86)	0.004

Abbreviations: ACE, angiotensin-converting enzyme; BMI, body mass index; CAD, coronary artery disease; CABG, coronary artery bypass graft; cm, centimeters; EF, ejection fraction; HFrEF, heart failure with reduced ejection Fraction (≤40%); Kg, kilogram; m, meter; PCI, percutaneous coronary intervention, Smoking, current smoker. Data are mean ± SD unless otherwise stated (*n* %).

**Table 2 jcm-12-07743-t002:** Cardiopulmonary exercise testing of participants.

Time	Characteristic	HfrEF	CAD	*p*-Value
Rest	VO_2_ (L/min)	0.333 (0.3–0.484)	0.285 (0.213–0.386)	0.680
	VO_2_ (mL/kg)	4.4 (3.2–6.5)	3.1 (2.6–4.4)	0.680
	RER	0.84 ± 0.05	0.84 ± 0.07	0.489
	VE (L/min)	15 ± 8	16 ± 10	0.642
	SpO_2_	97 ± 2	97 ± 2	0.899
	HR (bpm)	78 ± 12	76 ± 10	0.452
	SBP (mm Hg)	100 ± 19	112 ± 10	0.192
	DBP (mm Hg)	68 ± 9	72 ± 10	0.571
VT1	Workload (Watt)	75 (65–100)	90 (75–100)	0.202
	VO_2_ (L/min)	0.971 (0.901–1.541)	1.223 (1.03–1.35)	0.326
	VO_2_ (mL/kg·min)	13.1 (11.5–16.5)	14.2 (12.3–17.3)	0.978
	% VO_2_p (mL/kg·min)	76 (69–80)	69 (63–73)	0.160
	% VO_2_p predicted (mL/kg·min)	54 ± 17	55 ± 17	0.451
	RER	0.96 ± 0.05	0.97 ± 0.07	0.314
	VE (L/min)	46 ± 12	42 ± 11	0.181
	EqCO_2_	40 ± 9	35 ± 5	0.06
	PetCO_2_ (mm Hg)	34 ± 5	38 ± 4	0.06
	SpO_2_ (%)	97 ± 4	97 ± 2	0.8
	HR (bpm)	97 (91–103)	102 (90–108)	0.468
	SBP (mm Hg)	132 ± 36	133 ± 24	0.927
	DBP (mm Hg)	75 ± 19	76 ± 22	0.940
Peak	Workload (watt)	125 ± 40	156 ± 48	0.208
	VO_2_ (L/min)	1.6 ± 0.6	1.8 ± 0.6	0.428
	VO_2_ (mL/kg)	17.2 (16.2–21.1)	20.3 ± (19.8–25.4)	0.605
	%VO_2_ predicted	72 (61–77)	73 (62–87)	0.900
	RER	1.24 ± 0.11	1.21 ± 0.12	0.705
	VE (L/min)	80 ± 17	84 ± 18	0.644
	BR (%)	36 ± 15	35 ± 22	0.931
	SpO_2_ (%)	96 ± 4	96 ± 3	0.356
	HR (bpm)	126 ± 26	132 ± 18	0.484
	%Hrmax	78 ± 11	83 ± 9	0.218
	SBP (mm Hg)	166 ± 55	190 ± 26	0.169
	DBP (mm Hg)	82 ± 19	100 ± 31	0.113
Slope	Workload/VO_2_	9 ± 2	9 ± 1	0.690
	HR/VO_2_	3.2 ± 0.9	3.2 ± 1	0.906
	VE/VCO_2_	42 ± 11	35 ± 6	0.082
	HR recovery 1 min (bpm)	19 ± 9	20 ± 8	0.794
	HR recovery 2 min (bpm)	32 ± 10	32 ± 9	0.917

Values are mean ± SD. Abbreviations: BR: breathing reserve, CAD, coronary artery disease, DBP: diastolic blood pressure, EqCO_2_: carbon dioxide equivalent, HFrEF, heart failure with reduced ejection fraction (EF ≤ 40%), HR: heart rate, Peak: peak of exercise, PetCO_2_: end-tidal pressure of CO_2_, RER: respiratory exchange ratio, Rest: rest before Cpet, SBP: systolic blood pressure, SpO_2_: peripheral oxygen saturation, VE: ventilation, VO_2_: O_2_ uptake, VT1: anaerobic threshold.

**Table 3 jcm-12-07743-t003:** Hormone response after HIIT and resistance training methods.

Participant	Hormone	t0	t1	t2
	Growth hormone 3/7	0.08 (0.05–0.16)	2.93 (2.04–3.52)	1.62 (0.66–2.83)
HFrEF	Growth hormone 3x9	0.19 (0.07–0.44)	2.8 (1.71–6.35) **	1.68 (0.64–2.63)
	Cortisol 3/7	222 (190–326)	295 (236–432)	266 (246–428)
	Cortisol 3x9	277 (254–346)	389 (289–515)	374 (255–389)
	Growth hormone 3/7	0.24 (0.06–0.61)	3.42 (1.53–6.74) ***	1.79 (0.79–3.2)
CAD	Growth hormone 3x9	0.12 (0.07–0.44)	3.45 (1.11–4.67) **	1.37 (0.45–2.14)
	Cortisol 3/7	212 (148–243)	330 (174–396) *	361 (171–416)
	Cortisol 3x9	180 (128–271)	324 (262–371) *	336 (242–366)

Values are median and interquartile range. * indicates a difference between t0 and t1 at *p* < 0.05, ** *p* < 0.01, *** *p* < 0.001; CAD, coronary artery disease; HFrEF, heart failure with reduced ejection fraction (≤40%).

**Table 4 jcm-12-07743-t004:** Rating of perceived exertion and preference after 3/7 and 3x9 resistance training methods.

Participant	HFrEF		CAD			
RT Method	3x9	3/7	*p* Value	3x9	3/7	*p* Value
RPE	4.6 ± 3.1	5.2 ± 2.8	0.295	5.5 ± 2.6	6.7 ± 1.7	0.124
Enjoyment	8 (7.5–9)	9 (8–9)	0.053	7.6 ± 1.6	7.9 ± 1.7	0.720

Values are mean ± SD or median and interquartile range; CAD, coronary artery disease; HFrEF, heart failure with reduced ejection fraction (≤40%); RT method, resistance training method, RPE, rating of perceived exertion.

## Data Availability

Alexis Gillet had full access to all the data in the study and takes responsibility for the integrity of the data and the accuracy of the data analysis. Data can be obtained from the corresponding author.

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
