# Peer review of "Effects of High-Intensity Interval Training Using the 3/7 Resistance Training Method on Metabolic Stress in People with Heart Failure and Coronary Artery Disease: A Randomized Cross-Over Study"

_jcm, 2023, doi:10.3390/jcm12247743_

Round 1
Reviewer 1 Report
Comments and Suggestions for Authors
Author Response
Reviewer 1:
Summary: The aim of this manuscript by Gillet et al. was to test the hypothesis that the 3/7 resistance training method induces more metabolic stress in individuals with heart failure with reduced ejection fraction (HFrEF) or coronary artery disease (CAD) than the 3X9 resistance training method, when preceded with high-intensity interval training (HIIT). The main contribution of this manuscript is that it adds to the literature by investigating the metabolic stress effects of an acute single bout of resistance training following HIIT using either the 3/7 method or 3/9 method in participants with HFrEF or CAD. The results showed that the 3/7 RT method induced more metabolic stress than the 3x9 method in people with CAD but not HFrEF. Strengths of this study are that it compares the effects of the two resistance training methods in participants with HFrEF and CAD using a randomized crossover trial design. In addition, a notable strength of this manuscript is how well the study was conducted overall and written.
R: Thank you for your insightful review and valuable feedback. We appreciate your recognition of our study's contribution to understanding metabolic stress (MS) in RT methods for CAD and HFrEF patients.
General comments: Overall, this study was generally well conducted and written. This manuscript adds to the literature by investigating the acute metabolic stress effects of a single bout each of two different resistance training methods, either the 3/7 method or 3/9 method following HIIT in participants with HFrEF or CAD. The findings from this study provide information that while the more traditional 3/9 method of resistance training, which has been shown to be effective in numerous populations, the 3/7 method is also effective at increasing metabolic stress and is more time efficient. Given that the 3/7 method is relatively new compared to the more traditional 3/9 method and time is often cited as a limiting factor for participating in exercise, this information is beneficial for those working with people in cardiac rehabilitation in the design and implementation of resistance training programs. I do have any major concerns or issues with this manuscript. However, a limitation inherent to the acute study design is what effects the 3/7 resistance training method chronically results in.
R: We fully agree and now added in the limitation section a that (line 357-358) “Whether the acute effects reported in this study will translate into long-term beneficial effects is unknown and will require further long-term studies” as you recommend. This critical aspect is a key focus for our upcoming Trial. (NCT05696990)
Specific comments:
General comment: Check reporting of p-values throughout because there are some inconsistencies/grammatical errors in the way some of p-values are written (e.g., line 24, line 25, line 26, line 186, line 202, line 204).
R: Thank you for pointing out the typographical errors in our manuscript. We have corrected them following your observations.
Formerly:
Page 1, line 24-27:)” In the CAD group, lactate increased more (6.99±2.37 vs 9.20±3.57 mmol/L,p=0.025), pH, tended to decrease more (7.29±0.06 vs 7.33±0.04, p=.060) and HCO3- decreased more (18.6±3.1 vs 21.1±2.5 mmol/L, p=.004) after 3/7 than 3X9 RT. In the HFrEF group, lactate, pH and HCO3- concentrations did not differ between RT methods (all p>.248).”
Page 5, line 185-186 : “Mean peak oxygen consumption was similar in both groups (HFrEF: 20.7±7.1 and CAD: 21.3±4.8 mL/min.Kg, p=.601)”
Page 6, line 202-204: “For the CAD group, pH was not affected by HIIT but decreased similarly after both RT modalities with a tendency for a lower pH after the 3/7 RT (p=0,0601). HCO3- fell after HIIT with no further change after both RT modalities. However, HCO3- was lower after 3/7 RT than 3x9 RT (p=0,0044).”
Now changed into:
Page 1 line 24-27: “In the CAD group, lactate increased more (6.99±2.37 vs 9.20±3.57 mmol/L,p=0.025), pH, tended to decrease more (7.29±0.06 vs 7.33±0.04, p=0.060) and HCO3- decreased more (18.6±3.1 vs 21.1±2.5 mmol/L, p=0.004) after 3/7 than 3X9 RT. In the HFrEF group, lactate, pH and HCO3- concentrations did not differ between RT methods (all p>0.248))
Page 5, line 190-191 : “Mean peak oxygen consumption was similar in both groups (HFrEF: 20.7±7.1 and CAD: 21.3±4.8 mL/min.Kg, p=0.601)”
Page 6, line 208-211: “For the CAD group, pH was not affected by HIIT but decreased similarly after both RT modalities with a tendency for a lower pH after the 3/7 RT (p=0.060). HCO3- fell after HIIT with no further change after both RT modalities. However, HCO3- was lower after 3/7 RT than 3x9 RT (p=0.004).”
Lines 239-240: Stating that the study “confirmed our hypothesis that the RT 3/7 method would induce more metabolic stress than the RT 3x9 method” is only partially true, as the authors go on to state “however this was only true for the CAD and not the HFrEF group”. It is recommended to restate/reword this sentence to more accurately reflect the findings of this study. Lines 342-351 (Limitations): I recommend including the inherent limitation of the acute study design and whether or not acute effects observed may translate to beneficial outcomes chronically, which could also be a future research recommendation.
R: We agree with your comment and already followed your recommendation, stated in “general comments” (see above), to take into account the beneficial outcomes chronically in limitations section. We have modified the link between results and the hypothesis when introducing the discussion.
Formerly:
Page 8, line 244-246: “This study confirmed our hypothesis that the RT 3/7 method would induce more metabolic stress than the RT 3x9 method, however this was only true for the CAD and not the HFrEF group”
Now changed into:
Page 8, line 244-246: “This study partially confirmed the hypothesis that the 3/7 RT method would induce more metabolic stress than the 3x9 method after a HIIT session, specifically in participants with CAD. However, this was not observed in the HFrEF group”
We thank the Reviewer for these positive comments. We hope that our study is now suitable for publication in the Journal of Clinical Medicine
Reviewer 2 Report
Comments and Suggestions for Authors
Question 1: Line 109 mentions a washout period of >48 hours, which is too broad a value. Is the duration of the washout period the same for everyone? If not, is there a statistical difference in the washout period duration between the HFrEfa group and the CAD group?Additionally, please include references for the washout period duration.
Question 2: Your research conclusion should be "The 3/7 RT method induced more metabolic stress than the 3x9 RT method after a HIIT session in people with CAD but not in those with HFrEF." This is because your research protocol involved conducting high-intensity interval training before resistance training, rather than resistance training alone.
Question 3: Line 123 mentions that each cartridge contained sensors to measure pH, pCO2, pO2, electrolytes (Na+, K+, Ca++, Cl-), glucose, and lactate. However, the research results for pCO2, electrolytes (Na+, K+, Ca++, Cl-), and glucose are not found in the subsequent text.
Question 4: All the indicators in the article should be clearly described in terms of the methods used for measurement, such as systolic blood pressure, diastolic blood pressure, and other parameters.
Question 5: Please briefly explain why the variable of alcohol consumption was not measured.
Question 6: The full term (abbreviation) should be used when it first appears. There are multiple instances in the article where the full term is not used. Please double-check and verify.
Question 7: The tables are not standardized: 1) Table 1 and Table 2 lack table headers, 2) There are punctuation errors in the table notes of Table 3, 3) Table 1, Table 2, and Table 4 lack "P<0.05" to indicate significant differences in the table notes, 4) In Table 2, "VE" is written in the table, while "Ve" is written in the table note; the capitalization should be consistent (similar inconsistency in capitalization exists for the same variable name in the article), 5) Adjust the column spacing of Table 4, 6) Since it is mentioned in line 92 that the study included male subjects, Table 1 does not need to show the number of male subjects included.
Question 8: Figure 1 is not clear.
Question 9: There are several instances of improper writing in the article, such as p=.004 in line 26; p=.601 in line 186; (p=0,0601) in line 202; (p=0,0044) in line 204.
Comments on the Quality of English LanguageModerate editing of English language required
Author Response
Reviewer 2:
Question 1: Line 109 mentions a washout period of >48 hours, which is too broad a value. Is the duration of the washout period the same for everyone? If not, is there a statistical difference in the washout period duration between the HFrEfa group and the CAD group?Additionally, please include references for the washout period duration.
R: Thank you for your comment. We agree with your observation and have incorporated the concept of the washout period into the article. Furthermore, we have added in the results section the time between the two measurements in each group, which was identical (HFrEF 9 ± 4 days; CAD 7 ± 4 days; p=0.140). Additionally, we have conducted a multivariate analysis, attached herewith, which does not indicate a correlation between the number of days between the two tests and the study variables. We did not find any evidence suggesting that a shorter time period could have altered the results. Please refer to the attached reference below.
Formerly:
Page 3, line 111-112: “… at the same time of day (between 1 and 3 pm) and separated by more than 48 hours.”
Now changed into:
Page 3, line 111-112: “…at the same time of day (between 1 and 3 pm) and separated with a minimum rest period of two days, although a 7 days rest was considered ideal [Caldwell]”
Page 5, line 178-188 : ” The number of days between the two tests was similar (7±4 for HFrEF and 9±4 days for CAD (p= 0.140)). »

REF: Caldwell LK, et al. Acute Floatation-REST Improves Perceived Recovery After a High-Intensity Resistance Exercise Stress in Trained Men. Med Sci Sports Exerc. 2022 Aug 1;54(8):1371-1381. doi: 10.1249/MSS.0000000000002906. Epub 2022 Apr 6. PMID: 35389942.
Question 2: Your research conclusion should be "The 3/7 RT method induced more metabolic stress than the 3x9 RT method after a HIIT session in people with CAD but not in those with HFrEF." This is because your research protocol involved conducting high-intensity interval training before resistance training, rather than resistance training alone.
R: We agree that the influence of HIIT cannot be excluded in the response. The objective was to get closer to what is done in cardiac rehabilitation centers by having both types of training (aerobic and resistance). We were also concerned that it would be unethical to eliminate one of these trainings modalities in cardiac rehabilitation.
Formerly:
This study partially confirmed the hypothesis that the 3/7 RT method would induce more metabolic stress than the 3x9 method, specifically in participants with CAD. However, this was not observed in the HFrEF group.
Now changed into:
This study partially confirmed the hypothesis that the 3/7 RT method would induce more metabolic stress than the 3x9 method after a HIIT session, specifically in participants with CAD. However, this was not observed in the HFrEF group.
Question 3: Line 123 mentions that each cartridge contained sensors to measure pH, pCO2, pO2, electrolytes (Na+, K+, Ca++, Cl-), glucose, and lactate. However, the research results for pCO2, electrolytes (Na+, K+, Ca++, Cl-), and glucose are not found in the subsequent text.
R: The Reviewer is right and we strongly apologize for this oversight. The measurements for pCO2, electrolytes (Na+, K+, Ca++, Cl-), and glucose, although initially mentioned, were considered not relevant to the focus of the paper, and hence were not analyzed. We have removed these items for clarity.
Formerly:
Page 3, line 122-123: “….each cartridge contained sensors to measure pH, pCO2, pO2, electrolytes (Na+, K+, Ca++, Cl-), glucose and lactate”
Page 4, line 162-163: “The effects of method and time on pH, lactate, HCO3-, and pCO2 concentrations were evaluated using linear mixed-effects models”
Now changed into:
Page 3, line 126-127: “….each cartridge contained sensors to measure pH, HCO3- and lactate”
Page 4, line 171-172: “The effects of method and time on pH, lactate, and HCO3-, concentrations were evaluated using linear mixed-effects models”
Question 4: All the indicators in the article should be clearly described in terms of the methods used for measurement, such as systolic blood pressure, diastolic blood pressure, and other parameters.
R: Similar to the variables such as heart rate or systolic and diastolic blood pressure, the measurements were done during the CPET at rest, VT1 and Peak exercise. We have clarified this point.
We have included a detailed description of the CPET in the supplementary files. Recognizing the importance of this test in demonstrating exercise capacity, and considering that it was performed only once, we thought that it was not mandatory to incorporate CPET into the main text.
Now changed into:
Page 3, line 105-107: “During CPET, hemodynamic variables such as heart rate and blood pressure were measured at rest and throughout the effort, anaerobic threshold was determined by the V-slope method and the VE/VCO2 slope utile the second ventilatory threshold [Chaumontl].”
Supplemental files :
“Cardiopulmonary Exercise Testing (CPET):
CPET was performed using a cycle ergometer (Ergoselect II 1200; Ergoline, Bitz, Germany) with a step-by-step increase in workload. The rate of work increment (W·min−1) was personalized based on expected exercise tolerance and resting functional data. HR was derived a standard ECG signal analysis and blood pressure was obtained with an automatic pneumatic sphygmomanometer. Oxygen uptake (VO2), carbon dioxide output (VCO2), and ventilation (VE) were measured breath by breath through a facial mask and analyzed every 8 seconds using a metabolic system (Exp’Air®, Medisoft, Dinant, Belgium) that was calibrated with room air and standardized gas. CPET was considered maximal when two of the following criteria were met: Oxygen uptake increase less than 100 mL/min with increasing workload, respiratory exchange ratio above 1.10, attainment of age-predicted maximal heart rate, ventilator reserve less than 15%, and inability to maintain pedal rate above 50 revolutions per minute. Ventilatory threshold (VT1) was determined with the Vslope and Ventilatory efficiency was determined by the VE/VCO2 slope using linear regression analysis up to the ventilatory compensation point (i.e., secondary ventilatory threshold). (Chaumont 2023)
Chaumont M, et al. How Does the Method Used to Measure the VE/VCO2 Slope Affect Its Value? A Cross-Sectional and Retrospective Cohort Study. Healthcare (Basel). 2023 Apr 30;11(9):1292. doi: 10.3390/healthcare11091292. PMID: 37174834; PMCID: PMC10178610.
Question 5: Please briefly explain why the variable of alcohol consumption was not measured.
R: In our study, the measurement of alcohol consumption was not deemed essential. The primary focus of the research was on the direct physiological responses to training during cardiac rehabilitation, such as changes in pH, and Lactate. Our study aimed to isolate and observe specific RT outcomes in a cross over study, minimizing confounding variables that could arise from lifestyle choices of patients like alcohol consumption. Moreover, we believed that alcohol consumption is a lifestyle factor that might not directly correlate with the immediate effects of physical training (Levitt 2020). Last and importantly, all patients are followed and scrutinized by a professional dietician outside of our work, which strongly discourages alcohol consumption (Visseren 2021). This recommendation was provided repeatedly to all patients.
Levitt DE, et al. Alcohol After Resistance Exercise Does Not Affect Muscle Power Recovery. J Strength Cond Res. 2020 Jul;34(7):1938-1944. doi: 10.1519/JSC.0000000000002455. PMID: 29385007.
Visseren FLJ, et al. 2021 ESC Guidelines on cardiovascular disease prevention in clinical practice. Eur Heart J 42: 3227–3337, 2021. doi: 10.1093/eurheartj/ehab484.
Question 6: The full term (abbreviation) should be used when it first appears. There are multiple instances in the article where the full term is not used. Please double-check and verify.
R: Thank you for pointing out the typographical errors in our manuscript. We have corrected them following your observations.
Formerly:
Page 1 line 42-43 : “….and quality of life and reduces hospitalization rates in both people with heart failure with reduced ejection fraction”
Page 8 line 258-259 : “One reason for this difference could be the chronic myopathy often present in the case of heart failure [46].”
Page 9 line 275-276 “….with muscle abnormalities commonly seen in highly deconditioned people with heart failure [50,51].
Page 9 line 279-281: “Iron deficiency could also explain different metabolic stress responses during exercise in the muscle cells of people with heart failure. People with heart failure and iron-deficiency…”
Page 9 line 284-285: “…although his exercise capacity was lower than that of the other participants in the HrrEF group (VO2p 17 ml/kg.min),”
Page 9 line 293-294: “…altered by cardiac output in people with heart failure [53].”
Page 9 line 314: “…are contradictory in heart failure [59,60]”
Page 9 line 316: “…in people with heart failure because they finally…”
Page 9 line 328-331 : “Neurohumoral dysfunction is a key feature of heart failure [2] but very little information is available regarding the pituitary gland secretion of growth hormone in heart failure. Importantly, it has been shown recently that about a third of people with heart failure….”
Now changed into:
Page 1 line 42-43: “…and quality of life and reduces hospitalization rates in both people with heart failure with reduced ejection fraction (HFrEF)”
Page 9 line 264-265: “One reason for this difference could be the chronic myopathy often present in the case of HFrEF [48].”
Page 9 line 279-280: “…abnormalities commonly seen in highly deconditioned people with HFrEF [50,51].”
Page 9 line 285-286: “Iron deficiency could also explain different metabolic stress responses during exercise in the muscle cells of people with HFrEF. People with HFrEF and iron-deficiency have lower peak muscle strength. »
Page 9 line 290-291: “…although his exercise capacity was lower than that of the other participants in the HFrEF group (VO2p 17 ml/kg.min),”
Page 9 line 299-300: “…altered by cardiac output in people with HFrEF [53].”
Page 9 line 320: “…are contradictory in HFrEF [59,60]”
Page 9 line 322: “…in people with HFrEF because they finally…”
Page 9 line 334-336 “Neurohumoral dysfunction is a key feature of HFrEF [2] but very little information is available regarding the pituitary gland secretion of growth hormone in HFrEF. Importantly, it has been shown recently that about a third of people with HFrEF…”
Question 7: The tables are not standardized: 1) Table 1 and Table 2 lack table headers, 2) There are punctuation errors in the table notes of Table 3, 3) Table 1, Table 2, and Table 4 lack "P<0.05" to indicate significant differences in the table notes, 4) In Table 2, "VE" is written in the table, while "Ve" is written in the table note; the capitalization should be consistent (similar inconsistency in capitalization exists for the same variable name in the article), 5) Adjust the column spacing of Table 4, 6) Since it is mentioned in line 92 that the study included male subjects, Table 1 does not need to show the number of male subjects included.
R: Thank you for pointing out the typographical errors in our manuscript.
Question 8: Figure 1 is not clear.
R: We have modified Figure 1 and hope that it is now clearer. Additionally, the flow chart is available for the allocation of groups.
Question 9: There are several instances of improper writing in the article, such as p=.004 in line 26; p=.601 in line 186; (p=0,0601) in line 202; (p=0,0044) in line 204.
R: Thank you for pointing out the typographical errors in our manuscript. We have corrected them following your observations.
Formerly:
Page 1, line 24-27:)” In the CAD group, lactate increased more (6.99±2.37 vs 9.20±3.57 mmol/L,p=0.025), pH, tended to decrease more (7.29±0.06 vs 7.33±0.04, p=.060) and HCO3- decreased more (18.6±3.1 vs 21.1±2.5 mmol/L, p=.004) after 3/7 than 3X9 RT. In the HFrEF group, lactate, pH and HCO3- concentrations did not differ between RT methods (all p>.248).”
Page 5, line 185-186 : “Mean peak oxygen consumption was similar in both groups (HFrEF: 20.7±7.1 and CAD: 21.3±4.8 mL/min.Kg, p=.601)”
Page 6, line 202-204: “For the CAD group, pH was not affected by HIIT but decreased similarly after both RT modalities with a tendency for a lower pH after the 3/7 RT (p=0,0601). HCO3- fell after HIIT with no further change after both RT modalities. However, HCO3- was lower after 3/7 RT than 3x9 RT (p=0,0044).”
Now changed into:
Page 1 line 24-27: “In the CAD group, lactate increased more (6.99±2.37 vs 9.20±3.57 mmol/L,p=0.025), pH, tended to decrease more (7.29±0.06 vs 7.33±0.04, p=0.060) and HCO3- decreased more (18.6±3.1 vs 21.1±2.5 mmol/L, p=0.004) after 3/7 than 3X9 RT. In the HFrEF group, lactate, pH and HCO3- concentrations did not differ between RT methods (all p>0.248))
Page 5, line 191-193 : “Mean peak oxygen consumption was similar in both groups (HFrEF: 20.7±7.1 and CAD: 21.3±4.8 mL/min.Kg, p=0.601)”
Page 6, line 211-215: “For the CAD group, pH was not affected by HIIT but decreased similarly after both RT modalities with a tendency for a lower pH after the 3/7 RT (p=0.060). HCO3- fell after HIIT with no further change after both RT modalities. However, HCO3- was lower after 3/7 RT than 3x9 RT (p=0.004).
We thank the Reviewer for these positive, insightful and very constructive comments which we believe have considerably improved our manuscript. We hope that our study is now suitable for publication in the Journal of Clinical Medicine.
Round 2
Reviewer 2 Report
Comments and Suggestions for Authors
Thank you for the revisions. I have no further suggestions. Congratulations!